# Effects of Guangzhou seasonal climate change on the development of *Aedes albopictus* and its susceptibility to DENV-2

**Shanshan Wu**[1], **Yulan He**[1], **Yong Wei**[1], **Peiyang Fan**[1], **Weigui Ni**[1], **Daibin Zhong**[2], **Guofa Zhou**[2], **Xueli Zheng**[1] *

**1** Department of Pathogen Biology, School of Public Health, Southern Medical University, Guangzhou, China, **2** Program in Public Health, College of Health Sciences, University of California, Irvine, CA, United States of America

* zhengxueli2001@126.com

**Data Availability Statement:** All relevant data are within the manuscript and its Supporting Information files.

## Abstract

The susceptibility of Asian tiger mosquitoes to DENV-2 in different seasons was observed in simulated field environments as a reference to design dengue fever control strategies in Guangzhou. The life table experiments of mosquitoes in four seasons were carried out in the field. The susceptibility of *Ae. albopictus* to dengue virus was observed in both environments in Guangzhou in summer and winter. *Ae. albopictus* was infected with dengue virus by oral feeding. On day 7 and 14 after infection, the viral load in the head, ovary, and midgut of the mosquito was detected using real-time fluorescent quantitative PCR. Immune-associated gene expression in infected mosquitoes was performed using quantitative real-time reverse transcriptase PCR. The hatching rate and pupation rate of *Ae. albopictus* larvae in different seasons differed significantly. The winter hatching rate of larvae was lower than that in summer, and the incubation time was longer than in summer. In the winter field environment, *Ae. albopictus* still underwent basic growth and development processes. Mosquitoes in the simulated field environment were more susceptible to DENV-2 than those in the simulated laboratory environment. In the midgut, viral RNA levels on day 7 in summer were higher than those on day 7 in winter (F = 14.459, P = 0.01); ovarian viral RNA levels on day 7 in summer were higher than those on day 7 in winter (F = 8.656, P < 0.001), but there was no significant difference in the viral load at other time points (P > 0.05). Dicer-2 mRNA expression on day 7 in winter was 4.071 times than that on day 7 in summer: the viral load and Dicer-2 expression correlated moderately. *Ae. albopictus* could still develop and transmit dengue virus in winter in Guangzhou. Mosquitoes under simulated field conditions were more susceptible to DENV-2 than those under simulated laboratory conditions.

## Introduction

With the acceleration of climate warming, globalization, and urbanization, the epidemic scope of dengue fever is expanding worldwide [1]. Dengue fever is a worldwide public health

**Funding:** This work was supported by the Natural Science Foundation of China (No. 31630011), the Natural Science Foundation of Guangdong Province (No. 2017A030313625), and the Science and Technology Planning Project of Guangzhou (No. 201804020084). The funders had no role in study design, data collection and analysis, decision to publish, or preparation of the manuscript.

**Competing interests:** The authors have declared that no competing interests exist.

concern, with approximately 390 million people affected each year and 4 billion considered at risk [2]. A broad spectrum of clinical manifestations can be encountered, usually ranging from asymptomatic infections to mild-febrile illness. However, severe forms of dengue can involve hemorrhagic manifestations, sometimes with a fatal outcome [3]. No specific antiviral treatment is available and dengue vaccines require improvement [4]. Dengue fever is a mosquito-borne virus that has been prevalent for a long time in China. There have been many outbreaks in Guangdong, Hainan, and Zhejiang. In 2014, there were 45203 dengue cases and 6 deaths in Guangdong Province, of which 99.8% were local cases and 0.2% were imported cases, accounting for 96% of the total cases in China [5, 6]. The epidemic of dengue fever is seasonal. From 2005 to 2014, the peak of dengue fever cases occurred from July to November each year (Summer and Autumn) [7].

Guangzhou, the largest city in southern China, has been the epidemic center of dengue fever in China since the 1990s. In Guangzhou, *Aedes albopictus* (*Ae. albopictus*) is the only vector of dengue fever. The annual average temperature in Guangzhou is 21.6˚C, however, winter temperatures can be below 10˚ C. The annual rainfall is 1980 mm, making the climate in Guangzhou very suitable for the growth and reproduction of *Ae. albopictus* [8].

Many components of the dengue transmission cycle and the vector life cycle are temperature-dependent [9–11]. Environmental temperature is one of the most important abiotic factors affecting insect physiology, behavior, ecology, and even survival. *Ae. albopictus* is a typical temperature-sensitive insect, whose population is affected by seasonal temperature fluctuations. Therefore, observational data of *Ae. albopictus* development under natural conditions play an important role in monitoring vector population expansion, dengue virus transmission, and disease prevention [12–14].

In ecological research, a life table is often used to reflect the survival and death process of a population simply and intuitively [15]. The ecological habits and population size of *Ae. albopictus* are important factors for the transmission of dengue virus, and the life scale is an important means to understand the population dynamics of mosquitoes. Temperature is one of the important observation factors in life scale experiments. Many studies have explored the growth and development of *Ae. albopictus* at different temperatures; however, most experiments were carried out under constant temperature [16–24]. Yang et al., reported that semi-field life-table studies of *Ae. albopictus* (Diptera: Culicidae) in Guangzhou. *Ae. albopictus* larvae could develop and emerge and the adults could survive and produce eggs in early winter in Guangzhou. The major impact of changes in ambient temperature, relative humidity, and light intensity was on the egg hatching rates, adult survival time, and egg mass production, rather than on pupation or adult emergence rates [13].

Following ingestion via a bloodmeal, the arbovirus must first infect the midgut epithelial cells of the vector. Presumably, virions interact with receptors on midgut epithelial cells and penetrate the cells. Uncoating, transcription, and translation of the virus genome is followed by virion maturation. Then, infectious virions must disseminate from the midgut epithelium and infect secondary target organs. If the arbovirus is blocked at early stages of midgut infection, this is considered a midgut infection barrier (MIB). If infectious virions do not disseminate to hemoceles, this is considered a midgut escape barrier (MEB) [25, 26]. The vector ability of mosquitoes is the ability of virus to infect, proliferate, and transmit to other hosts after being inhaled by the arthropods [25–27]. Before the virus is transmitted to humans, it needs to propagate in the mosquito body for a period of time before it can be infectious. This period is called the external incubation period (EIP).

The measure of how efficiently an insect vector can transmit a pathogen is known as vector competence (VC) [28]. It is determined by a combination of environmental and genetic factors, such as temperature, mosquito nutrition, and viral and mosquito genotypes [25, 29–31].

The innate immune response of the mosquito is a key determinant for successful transmission of mosquito-borne viruses. Viral infection triggers the activation of innate immunity pathways in mosquitoes, including the RNA interference (RNAi) pathway, the Janus kinase-signal transducer and activator of transcription (JAK-STAT) pathway, the Toll pathway, and the immune deficiency (Imd) pathway [32–38], which leads to the transcription of genes responsible for antiviral responses. Rel1, Rel2, Dicer-2, and STAT are the key factors in the Toll, Imd, RNAi, and JAK STAT pathways, respectively [38–43], and these factors have shown effective resistance against viral infections in some mosquito species [44, 45]. These pathways can limit virus replication, and higher basal expression of immunity-related genes also confers increased resistance against the virus [46]. However, there is evidence of virus driven downregulation of immune-related gene expression, possibly as an adaptation to evade immune responses and assist viral survival [47–49].

Viral dynamics within the mosquito also depend upon temperature [50–54]. The arboviral extrinsic incubation period is strongly affected by temperature [55–58]. Liu et al., reported that temperature was an important factor the affects the ability of DENV-2 to infect the *Ae. albopictus* vector (18–32˚C). The higher the temperature, the faster the virus proliferates in *Ae. albopictus*, the easier it breaks through the midgut barrier, and the shorter the time it takes to spread to the ovary and salivary gland [56]. However, other studies have shown that the EIP of dengue virus in *Ae. aegypti in vivo* and *in vitro* is shortened under conditions of low temperature and large temperature fluctuations [53, 59, 60]. The environmental temperature under natural conditions does not remain constant, but oscillates between a minimum at night and a maximum during daytime. Results from studies using realistic fluctuating temperature profiles support the notion that fluctuating temperatures might alter estimates of both life history traits and the vector competence of mosquitoes, with the magnitude of the diurnal temperature range (DTR) being associated with the degree of response observed [53, 61, 62].

To date, the study of dengue virus from the proliferation and transmission of Asian tiger mosquitoes in the natural environment in the field has been rarely reported. In the real field environment, the temperature fluctuation is relatively large. In addition, humidity and other meteorological factors are also constantly changing. Therefore, the susceptibility and transmission ability of *Ae. albopictus* to dengue virus under constant temperature and humidity in the laboratory might be quite different from the actual situation in the field. Exploring the ability of *Ae. albopictus* to transmit dengue virus under simulated field conditions will clarify the transmission of dengue virus by *Ae. albopictus* under natural conditions, which has guiding significance for the prevention and control of dengue. The aim of the present study was to explore the effects of climate change in different seasons on Asian tiger mosquito development in Guangzhou. The susceptibility of Asian tiger mosquitoes to DENV-2 in different seasons was observed using simulated field environment conditions to provide a reference for the design of dengue fever control strategies in Guangzhou.

## Materials and methods

### Experiment flow chart

**Ethics approval and consent to participate.** No specific permits were required for the described field studies. For mosquito collection in residential areas, oral consent was obtained from field owners in each location. The use of mice in mosquito blood-feeding was performed in strict accordance with the recommendations in the Guide for the Care and Use of Laboratory Animals of the National Institutes of Health and the guidelines of Southern Medical University on the experimental use of mice. All the animals were handled according to approved

institutional animal care and use committee (IACUC) protocols (#2017–005) of Southern Medical University.

**Test mosquitoes.**  There were two sources of mosquitoes, one was a wild strain (Guangzhou strain) and the other was laboratory strain. Field strain *Ae. albopictus* larvae were collected in May 2018 from multiple (>10) breeding habitats in two residential areas in Baiyun District of Guangzhou, Guangdong province. Field collected larvae from different habitats were mixed by putting larvae from different habitats into the same bucket. Larvae were brought back to semi-field setting and reared in microcosms where life-table experiments were conducted. Emerged adults were allowed mating freely. This mix will reduce the bias due to differences in larval source and inbreeding [13]. Mosquitoes were reared until F3 eggs in the field condition. F3 eggs were used for first round of life-table experiments for three reasons, to allow for get enough eggs within one day, to allow for field mosquitoes to adapt the new environment and mouse blood. We did not observe bottlenecks or significant loss of mosquitoes during this process [13]. The laboratory strains of *Ae. albopictus* was reared under standard laboratory conditions since 1981.

**Semi-life scale experimental methods.**  The experiment time was divided into four seasons, and the experiment time in spring was from April 2, 2019 to April 22, 2019; from June 4, 2019 to June 20, 2019 in summer; from October 27, 2019 to November 6, 2019 in autumn; and from December 30, 2019 to January 27, 2020 in winter. In each season, the life table experiments of the two mosquito strains were started at the same time under the two experimental environments. The experimental group was divided into the laboratory environment control group and the simulated field environment group. Based on a previous study from our research group [13], another simulation experiment in summer and winter was performed, in which the experiment time of larvae in summer was from July 17, 2018 to July 30, 2018; and the experiment time of the field group larvae in winter was from January 18 to March 14, 2019. Experiments were conducted in campus of Southern Medical University. The site was the same as previously reported [13].

In brief, for each experiment, 200 eggs (2 days old) were placed in a stainless steel dish (32.5 cm × 26.5 cm × 6.5 cm) with 2 L of tap water (dechlorinated) and stored overnight. Four replicates were used for all settings and months. The egg hatching, larval development, death, pupation, and eclosion of *Ae. albopictus* were observed. The surviving larvae were counted, and their stage of development was recorded at about 18:00 hours every day. The pupae were counted and removed daily. The larvae were fed with Inch-Gold® turtle food every day, at an average of about 0.1 g per 100 larvae per day. Water levels in the dishes were checked daily and maintained by adding tap water stored overnight as needed. Water temperature and light intensity were measured using HOBO® data loggers (Onset, Bourne, MA, USA) [13].

Newly emerged adults were used in the life-table studies with protocols similar to that described in a previous study [13]. Briefly, 30 female and 30 male adult mosquitoes within 24 h post-emergence were placed in 30 × 30 cm mosquito cage. The mosquitoes were provided with 10% glucose daily, and every three days, a mouse was placed in each cage for approximately 4 hours to allow the mosquitoes to blood-feed, The mice were confined to a wooden board. These mice were purchased from the animal experimental center of Southern Medical University. Every three days, the body temperature, body hair shape, exercise status, and other clinical signs of the mice were recorded for monitoring. The cages were examined daily to count the number of surviving and dead mosquitoes, and the dead mosquitoes were removed. The eggs laid in each cage were counted daily. The water temperature and light intensity were measured using HOBO® data loggers [13]. The survival and fecundity of the adult mosquitoes were observed at about 18:00 hours every day.

The field environment was set in the garden of Baiyun District, Guangzhou. The garden was closed and managed by special personnel to avoid interference by irrelevant human factors. The environment for raising *Ae. albopictus* in the field comprised mosquito nets placed in the pavilions with sunshade roofs and ventilation around. The plants in the garden are luxuriant, which was conducive to the breeding of *Ae. albopictus* [13]. The laboratory environment was set up in the standard insect breeding room. The laboratory conditions were temperature: $27 \pm 1°C$, humidity: 70–80%, light cycle, day:night = 14h:10 h.

**Expanded culture of experimental mosquitoes.** In June 2019, *Ae. albopictus* larvae were captured in several breeding sites in Guangzhou. The larvae were fed with small fish diet and the adults were fed with 10% glucose solution. Two groups of experiments were carried out in the field and laboratory. The mosquitoes were provided with 10% glucose daily, and, every three days, a mouse was placed in each cage for approximately 4 hours to blood-feed the mosquitoes, the mice were confined to a wooden board, in order to oviposition of female mosquitoes. In July 2019, eggs of *Ae. albopictus* from Guangzhou were released in both the field and laboratory environments.

**Enrichment of DENV-2 in C6/36 cells.** Dengue virus 2 (New Guinea C, GenBank accession number: AF038403.1) was provided by the Key Laboratory of Tropical Disease Control of Sun Yat-sen University (Guangzhou, China). Mosquito C6/36 cells were cultured in Roswell Park Memorial Institute (RPMI)-1640 medium supplemented with 10% heat inactivated fetal bovine serum (FBS) and maintained at 28°C. Cells were grown in a 75-cm$^2$ culture flask and inoculated with DENV-2 at a multiplicity of infection (MOI) of 1. After gentle shaking for 15 min, the culture flask was incubated at 37°C and 5% $CO_2$ for 2 days until obvious cytopathic effects were observed, Generally, the culture temperature of C6/36 cells is 28°C, but for the benefit of virus proliferation, we adopt the culture temperature of 37°C [56]. The supernatant was harvested after centrifugation at $1,500 \times g$ for 5 min, separated into 0.5-mL aliquots, and frozen at −80°C. The Dengue virus 2 titer was determined using the 50% tissue culture infective dose (TCID50) method [63].

**Field environment simulation of infected mosquitoes orally with DENV-2.** Adult mosquitoes at 4–6 days after eclosion were placed at −20°C for low temperature anesthesia for 40 s. After observing that the mosquito could not fly, the mosquito was placed a glass dish on ice. Anesthetized female mosquitos were placed in a 3000 ml round plastic box and covered with gauze net.

Two climate chambers were used to simulate the field environment and the standard laboratory environment. Simulation of the field environment was set up using 24 time periods in a day, and the hourly temperature, humidity, and light data were taken from the monitoring data of the first 24 hours in the field and used for simulation. The simulation of the field environment temperature, humidity, and illumination intensity time for summer used data from July 4, 2019 to July 18, 2019; and for winter, the data was taken from December 9, 2019 to December 22, 2019. The laboratory environment experiment comprised conditions of a constant temperature of 28°C, a relative humidity of 70%, and a day:night light cycle of 14:10 h.

The Guangzhou strain female mosquitoes were put into the climate boxes. The female mosquitoes in the field environment experimental group were put into the climate box simulating the field environment, and the Guangzhou strain female mosquitoes in the laboratory environment experimental group were put into the climate box simulating the laboratory environment.

**Oral infection with DENV-2.** The mosquitoes were infected with the virus via the oral route. Mosquito infection was conducted in a Biological Safety Level 2 laboratory. Two days before infection, frozen DENV-2 stock was passaged once more through C6/36 cells. The titer of the fresh virus was 7.375–7.875 log10 TCID50/mL. The DENV-2 supernatant was collected

and mixed with defibrinated sheep blood at a ratio of 2:1. The blood meal was maintained at 37°C for 30 min and transferred into a Hemotek blood reservoir unit (Discovery Workshops, Lancashire, UK. Female mosquitoes in the above groups were glucose starved for 12–24 h and allowed to feed on the infectious blood meal for 1 h. The female mosquitoes were anesthetized at −20°C for 15 seconds, and the blood filled females were selected on ice in the new plastic boxes, 50 mosquitoes in each box, and 6 boxes of mosquitoes were tested in each group.

**Vector competence of *Ae. albopictus* for DENV-2.** The midgut, ovaries, and head of each mosquito from the above-mentioned environmental conditions were dissected at 7 and 14 days post-infection (dpi). The sample size collected from each group was 30 mosquitoes at both 7 and 14 dpi. The experiment was repeated independently three to five times.

The legs and wings of each mosquito were removed and washed three times in phosphate-buffered saline (PBS). Disposable insect microneedles were used to separate the midgut, ovaries, and head of each mosquito under an anatomical lens. Tissues were washed three times in PBS droplets and then transferred to 50 µL of TRIzol (Ambion, Life Technologies, Carlsbad, CA, USA) in 1.5-mL Eppendorf tubes. Total RNA was extracted according to the TRIzol manufacturer's protocol. cDNA was synthesized using a DENV-2-specific primer (5′– TGGTCTTTCCCAGCGTCAAT–3′), and the recommendations of the GoScript™ Reverse Transcription System (Promega, Madison, WI, USA) were followed.

Polymerase chain reaction (PCR) was used to detect DENV-2 in tissues. A pair of primers was synthesized as described previously (forward primer: 5′–TCAATATGCTGAAACGCGC GAGAAACCG–3′; reverse primer: 5′–TTGCACCAACAGTCAATGTCTTCAGGTTC–3′) [56, 57]. The target fragment comprised 511 bp, which was located in the partial capsid and membrane protein coding region. The total volume of the PCR reaction system was 25 µL, including 12.5 µL of Maxima Hot Start Green PCR Master Mix (Thermo Fisher Scientific Inc., Waltham, MA, USA), 0.5 µL of each primer (10 µM), 1 µL cDNA, and 10.5 µL RNase-free water. PCR reaction conditions were: 94°C for 3 min; followed by 35 cycles of 94°C for 30 s, 56°C for 30 s and 72°C for 1 min; and 72°C for 7 min. PCR products were identified using 1% agarose gel electrophoresis, ligated into vector pMD18-T (Takara, Dalian, China), and confirmed by sequencing. In comparison with the tissues from the control group, positive tissues were identified by detecting specific DENV-2 sequence.

The vector competences of the *Ae. albopictus* mosquitoes were evaluated by calculating the infection rate (IR), dissemination rate (DR), potential transmission rate (TR), as follows [64]:

IR = the number of positive mosquitoes/the total test number of mosquitoes,

DR = the number of positive ovaries/the number of positive midguts,

TR = the number of positive heads / the number of positive midguts.

**Quantification of DENV-2 in tissues.** The amount of DENV-2 in the positive tissues of mosquitoes was further detected by RT-PCR. The plasmid standard was constructed as described previously [65]. In brief, the 3'-UTR region of DENV-2 was amplified PCR using specific primers (forward primer: 5′–TCCCTTACAAATCGCAGCAAC–3′; reverse primer 5′–TGGTCTTTCCCAGCGTCAAT–3′). The fragment of 127 bp was cloned into vector pMD18-T and linearized using EcoRI. A standard curve was generated by analyzing serial 10 fold dilution of the plasmid. The qPCR reaction mixture contained 20 µL SYBR select master mix, 0.4 µL of each primer, 1 µL cDNA or the plasmid standard, and 8.2 µL RNase-free water per well. The reaction was performed in a 7500 Real-Time PCR System as follows: 50°C for 2 min, 95°C for 5 min; followed by 40 cycles of 95°C for 10 s, 60.6°C for 20 s, and 72°C for 20 s. The result of qPCR was ascertained using a non-template, negative control (mosquito infected with C6/36 cells) and a positive control (mosquito infected with DENV-2 at 0 dpi). The minimum detection threshold was 90.1 copies/reaction of DENV-2. Three wells were set for each

sample. The dissolution curves and cycle threshold (CT) values of samples and plasmids were compared, and the viral copy number of samples was calculated [56].

**Quantification of immune-associated gene expression levels in mosquitoes.** The expression levels of immune-associated genes in mosquitoes infected with DENV-2 were quantified by comparing the values with standard curves from recombinant plasmids containing gene fragments of *Rel1* and *Dicer2* according to our previous study [38]. qPCR reactions were performed using solutions with a final volume of 20 μL, containing 10 μL of SYBR® green master mix, 0.5 μM of each primer, 2 μL of template DNA, and 7 μL of RNase-free water. The primers for all the detected fragments were the same as those cited in our previous study [38]. The thermal cycling conditions were: 10 min at 95˚C; 40 cycles of 95˚C for 15 s, primer Tm for 30 s (Tm values for each primer pair are shown in S1 Table), and 72˚C for 30s.

**Data analysis.** Data analysis was completed in SPSS 20 (IBM Corp., Armonk, NY, USA). The emergence rate is calculated according to the proportion of survival to emergence of the first instar larvae of *Ae. albopictus*. The average development time of larvae is defined as the average duration from the first instar larvae to the emergence of adult mosquitoes [66]. The pupation rate and the eclosion rate were compared using Fisher's test. Single-factor analysis of variance was used to test the difference between multiple groups. The survival time of larvae and adults was analyzed using Kaplan–Meier survival analysis [67]. According to the theory of effective accumulated temperature, the relationship between the temperature and the time required for the growth and development of *Ae. albopictus* was established. The effective accumulated temperature was determined as follows: Each insect state can grow and develop above a certain temperature, that is, the threshold temperature of development. When the temperature is higher than the development threshold temperature, it is the effective temperature. The effective accumulated temperature is the sum of the temperatures required by an insect to complete a certain insect states [24, 68].

The effective accumulated temperature is a constant. The formula is as follows:

$$K = D \times (T - T_0) \qquad\qquad 1-1$$

where, K is the effective accumulated temperature, D is the average development duration, T is the daily average temperature, and $T_0$ is the development threshold temperature. If formula 1-1 is transformed and the development rate is B, B = 1 / D, the linear regression equation is as follows:

$$T = T_0 + K \times b \qquad\qquad 1-2$$

A chi squared test was used to compare the infection rates at different time points under the different experimental conditions, and the infection rate at the same time point under different experimental conditions. The number of virus copies was calculated, and logarithm transformation based on 10 was carried out. The amount of virus in different tissues were compared using analysis of variance. Pearson's chi squared test or Fisher exact probability method were used to analyze and compare the differences in IR, DR, and TR for different simulated wild environment and laboratory condition mosquitoes (Fisher's exact probability method was used when the minimum expected number was less than 5). $P < 0.05$ was considered statistically significant [56]. All analyses were conducted using SPSS 22.0 statistical software.

## Results

### Pupation rate and emergence rate of *Ae. albopictus* larvae in different experimental environments

In the field environment, the pupation rates of *Ae. albopictus* larvae of Guangzhou strain and laboratory strain were the lowest in spring, at 85.0% and 83.3% respectively; however, the pupation rates of *Ae. albopictus* larvae in summer, autumn, and winter were more than 95.0%. In the field, the emergence rates of the Guangzhou strain and laboratory strain were more than 80.0% all year round. The emergence rates of the two strains reached the highest in summer, at 93.3% and 95.0% respectively, followed by autumn (Fig 1A).

In the laboratory environment, the pupation rates of the Guangzhou strain and laboratory strain were the lowest in spring, at 80.0%, and the pupation rates of the Guangzhou strain in summer, autumn, and winter were more than 90.0%. The pupation rates of the laboratory

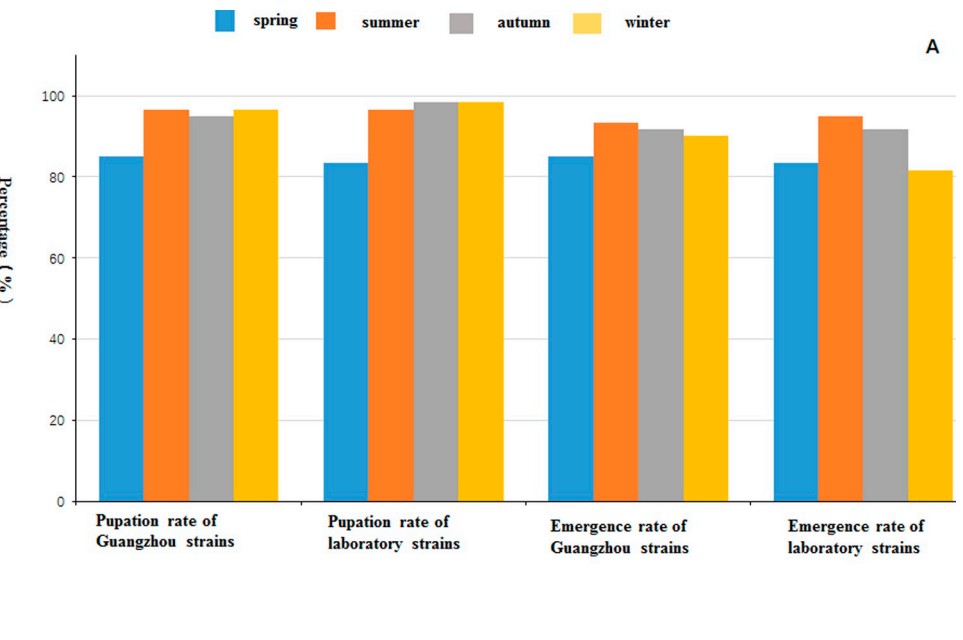

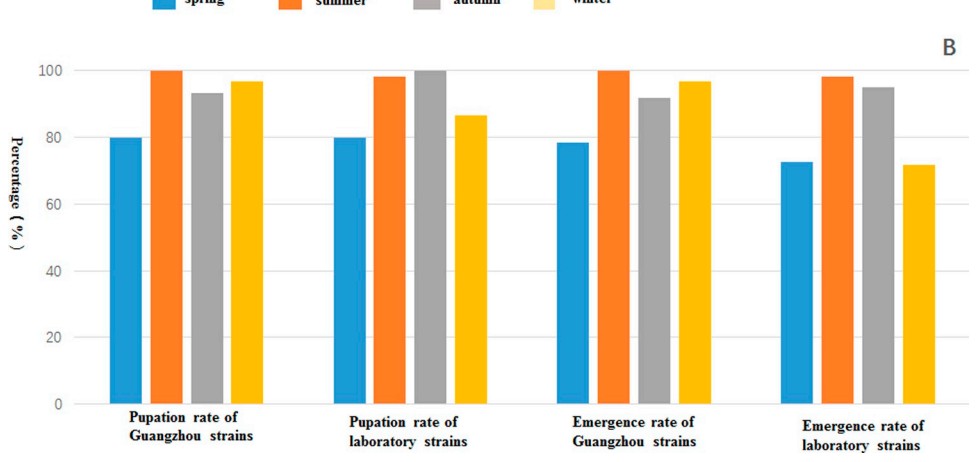

**Fig 1. Pupation rate and adult emergence rate of *Ae. albopictus* in wild conditions (A) and laboratory conditions (B).**

strains in summer and autumn were more than 95.0%, while that in winter was 86.7%. In the laboratory environment, the emergence rates of the Guangzhou strain in summer, autumn, and winter were more than 90.0%, and were the lowest in spring at 78.3%. The emergence rates of the laboratory strain in summer and autumn were more than 95.0%, and were lower in spring and winter, at 72.5% and 71.7%, respectively (Fig 1B).

Each mosquito strain in each experimental environment was divided into a group, i.e. the Guangzhou strain under the field environment, the laboratory strain under the field environment, the Guangzhou strain under laboratory conditions, and laboratory strain under laboratory conditions. The pupation rate of each group was compared longitudinally in time, and the difference in the pupation rate of each group in different seasons was analyzed using a chi squared test. Fisher's exact test was selected because there were less than five samples in each group. There was no significant difference in the pupation rate between the Guangzhou strain and the laboratory strain (P > 0.05). The P values of the pupation rate of two mosquito strains in the four seasons were 0.0002 and 0.00006, respectively, which were lower than the test level of 0.05. The pupation rate of each group of the Guangzhou and laboratory strains under laboratory conditions was significant different in the four seasons (P<0.05).

Table 1 showed that in the laboratory environment, the pupation rate of the Guangzhou strain in spring was lower than that in summer (P < 0.008). The pupation rate of the laboratory strain in spring was lower than that in summer and autumn (P < 0.008). The pupation rate of the laboratory strain in winter was lower than that in autumn (P < 0.008).

The larvae of *Ae. albopictus* (Guangzhou strain) in the laboratory environment group, the summer field environment group, and winter field environment group hatched. The hatching rate of *Ae. albopictus* larvae was different in the different environments and climates. The results showed that there was significant differences in hatchability between the summer field environment group and the winter field environment group ($\chi^2$ = 27.666, p < 0.05). The earliest hatching time of the eggs in the different environmental conditions was 1 day, and the latest start hatching time was in the winter field environment, in which the longest hatching time was 20 days. In the laboratory environment, the number of eggs hatched on the first day was the largest, and the average incubation time was the shortest (1.75 days). The average incubation time of eggs in the field environment group in winter was the longest, at 6.17 days (see S1 Table and Fig 2).

The larvae of *Ae. albopictus* developed into pupae in the laboratory environment group, summer field environment group, and winter field environment group in another simulated experiment. The difference of the pupation rate between the field environment group in summer and the field environment group in winter was statistically significant ($\chi^2$ = 061, P < 0.05) (S2 Table, Fig 2). The earliest pupation time of *Ae. albopictus* larvae under the different environmental conditions varied. The shortest pupation time of *Ae. albopictus* larvae in summer

**Table 1. P value for pairwise comparison of four seasonal pupation rates in different experimental groups.**

| Group | Laboratory environment | |
|---|---|---|
| | **Guangzhou strains** | **Lab strains** |
| Spring-summer | 0.0002 | 0.003 |
| Spring-autumn | 0.058 | 0.0004 |
| Spring-winter | 0.008 | 0.413 |
| Summer-autumn | 0.119 | 1.000 |
| Summer-winter | 0.496 | 0.032 |
| Autumn-winter | 0.679 | 0.006 |

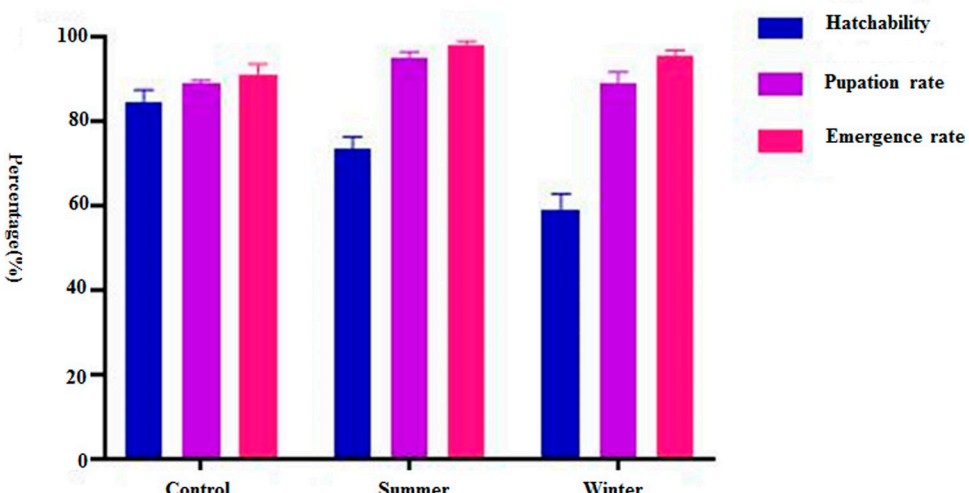

**Fig 2. Hatching rate, pupation rate, and emergence rate of *Ae. albopictus* under different environmental conditions.**

was 5 days. The longest pupation time was in the field environment in winter at 43 days (S2 Table).

The earliest emergence time of *Ae. albopictus* pupae was different under the different environmental conditions, in which the shortest emergence of pupae in the field environment group in summer was 7 days, the longest was in the field environment group in winter, at 25 days. The latest eclosion time of *Ae. albopictus* pupae was different under the different environmental conditions, in which the shortest eclosion time in the field environment group in summer was 12 days, and the longest eclosion time was in the winter field environment group, which lasted 49 days (S3 Table, Fig 2).

The survival of the Guangzhou strain and laboratory strain in the field environment is shown in Fig 3A, and that in the laboratory environment is shown in Fig 3B.

The survival of the Guangzhou strain in the laboratory environment, the laboratory strain in the laboratory environment, and each group in the four seasons was tested using the log rank test. The development of four groups of larvae in four seasons varied significantly (P < 0.05). The test level of the pairwise comparison was 0.008, as adjusted by the Bonferroni method. In the field, there was no significant difference in the development time between summer and autumn; however, there was a significant difference in the other seasons. In the laboratory environment, there was no significant difference in the development time between summer and winter; however, there was no significant difference in the development time between summer and autumn. There was a significant difference for the other seasons by pairwise comparison.

The development time of each stage of *Ae. albopictus* larvae in the field, and the development time of the Guangzhou strain and laboratory strain in the same season in the field environment, were basically the same. It took about 9 days from development to eclosion in summer and autumn, 15 days in spring, and 22 days in winter. In the laboratory environment, compared with the field environment, the time required for larval development to eclosion was varied less in the different seasons, and the median time required was between 8 and 11 days. The independent variable parameters of the model are shown in Table 2.

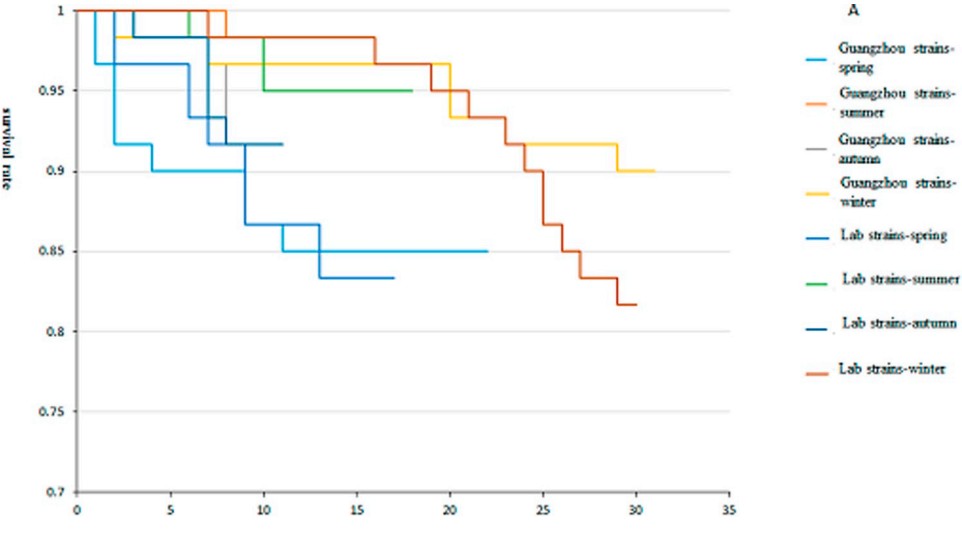

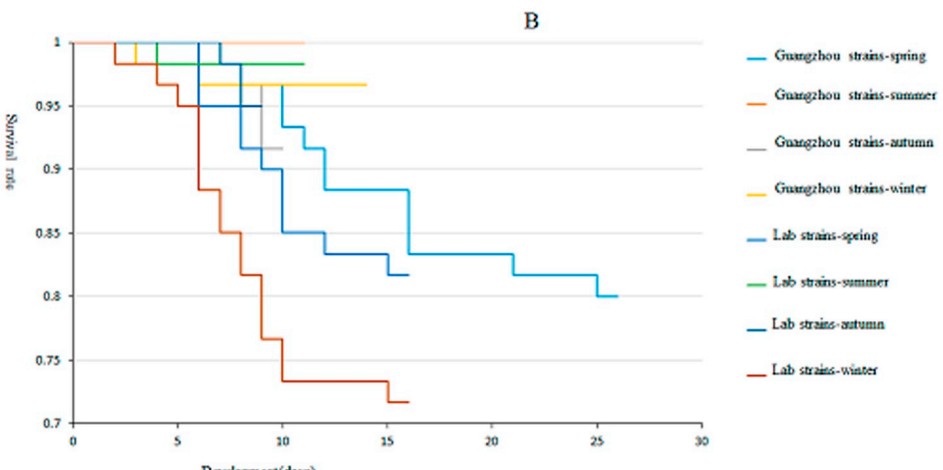

**Fig 3. Kaplan–Meier survival curves of *Ae. albopictus* larvae in the wild environment (A) and laboratory environment (B).**

**Table 2. P values for pairwise comparison of four seasonal emergence rates in different experimental groups.**

| Group | Field environment | Laboratory environment | |
|---|---|---|---|
| | **Lab strains** | **Guangzhou strain** | **Lab strains** |
| Spring-summer | 0.029 | 0.001 | 0.001 |
| Spring-autumn | 0.269 | 0.071 | 0.002 |
| Spring-winter | 1.000 | 0.004 | 1.000 |
| Summer-autumn | 0.439 | 0.057 | 0.619 |
| Summer-winter | 0.016 | 0.496 | 0.001 |
| Autumn-winter | 0.178 | 0.439 | 0.001 |

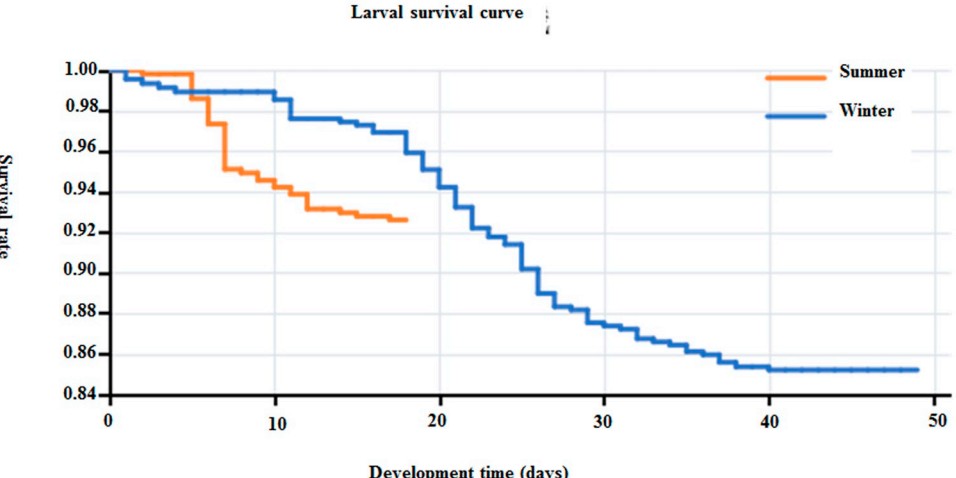

**Fig 4. Survival curve of *Ae. albopictus* larvae in summer and winter.**

Fig 4 shows the Kaplan-Meier curves of *Ae. albopictus* larvae in summer and winter. As shown in the Fig 4, the log rank test showed that the survival curves of the two groups were different, and the difference was statistically significant ($P < 0.05$).

The development of the larvae was less than 20 days in summer, but up to 50 days in winter.

## Adult development of *Ae. albopictus*

The chi squared test was used to test the eclosion rate in different seasons for the Guangzhou strain in the field environment, the laboratory strain in the field environment, the Guangzhou strain and the Laboratory strain in the laboratory environment. The P values were 0.535, 0.027, 0.001, and 0.001 respectively. The emergence rates of the three groups with P values less than 0.05 were compared in pairs. Bonferroni adjustment was 0.008, and the P values of the

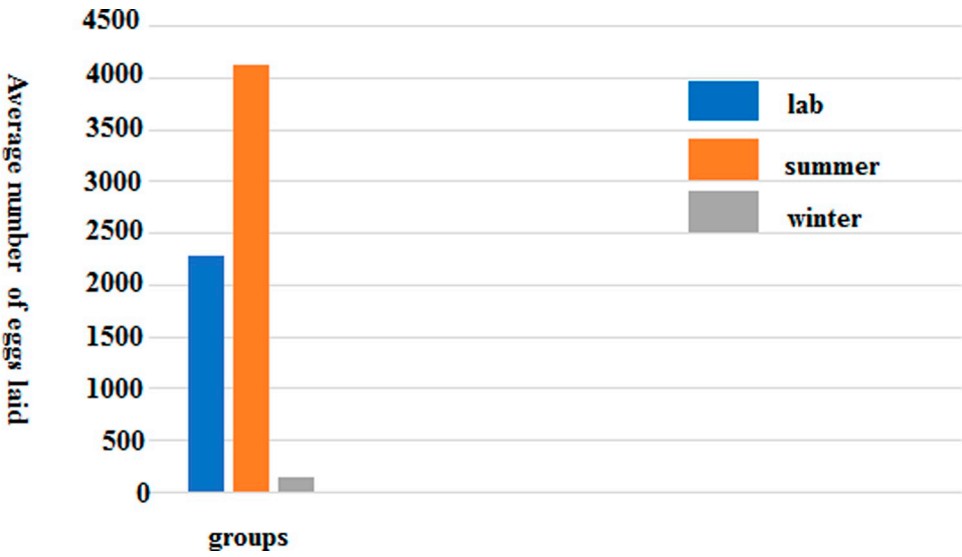

**Fig 5. Egg production of adult *Ae. albopictus*.**

pairwise comparisons are shown in Table 2. There was no significant difference in the emergence rate of the laboratory strains in the field over the four seasons (P > 0.008). In the laboratory, the emergence rate of the Guangzhou strain in spring and winter was lower than that in summer and autumn (P < 0.008).

Analysis of oviposition of mosquitoes showed that adult mosquitoes could normally suck blood and lay eggs in the laboratory and outdoor environment in summer, but sometimes they did not suck blood and lay eggs in field environment in winter. The group with the largest number of eggs laid was the summer field environment group, with a total of 10601 eggs, in which the average number of eggs laid by each adult mosquito was 176.68.

The group with the least number of eggs laid was the winter field environment group, total four groups, two of the groups did not lay eggs, but the total number of eggs laid in another two of the groups were 51 and 230, respectively (S4 Table, Fig 5).

The earliest oviposition was demonstrated by the adult mosquitoes in the laboratory, at 3 days. The latest oviposition was by the adult mosquitoes in the field in summer, which could still oviposit on the 54th day. According to the statistical analysis of the data of the laboratory control group, the summer field environment group and winter field environment group, the difference of average oviposition amount among the three groups was statistically significant (F = 43.288, P < 0.001). There were also significant differences in the average oviposition amount between two groups among the laboratory environment control group, summer field environment group, or winter field environment group (table Tukey's HSD test, P < 0.05).

The longest survival time of mosquitoes was different under the different environmental conditions. The longest survival time was the adult mosquitoes in the field environment group in summer, which could survive for 54 days. By contrast, the longest survival time was 36 days in the field environment in winter and 32 days in the laboratory environment (Fig 6).

The survival of adult *Ae. albopictus* was significantly different under different environmental conditions (F = 21.592, P < 0.001), in which the average survival time of adult mosquitoes was the longest in summer (average survival time = 22.88 ± 17.11 d (mean ± SD)), whereas the average survival time of adult mosquitoes was 10.47 ± 6.07 d in the laboratory and

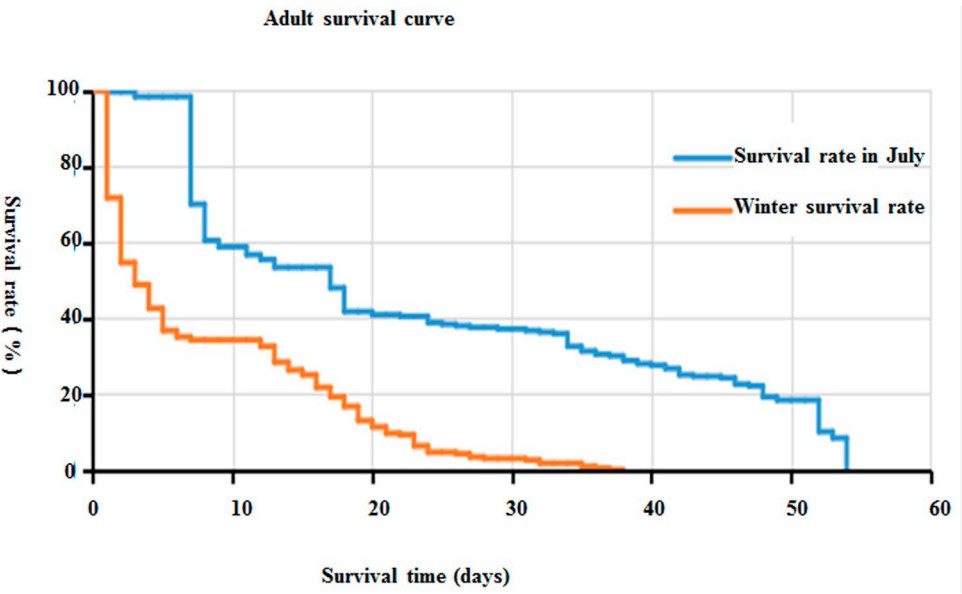

**Fig 6. Survival curve of adult *Ae albopictus*.**

**Table 3. Adult survival time of *Ae. albopictus* under different environmental conditions.**

| Experimental group | Female (days) | Male (days) |
|---|---|---|
| Laboratory | 11.20±6.16 A a | 9.73±6.00 A a |
| Summer experiment | 26.17±18.83 A b | 19.60±14.80 B b |
| Winter experiment | 13.23±12.34 A a | 6.93±7.48 B a |

** Tukey HSD test, numbers in the same row (column) not connected with the same capital (small) letters indicating significantly different from each other at level of 0.05

10.08 ± 10.60 d the field environment in winter. The survival time of female adults was longer than that of male adults (F = 8.291, P < 0.05). There was a significant difference in survival time between male and female *Ae. albopictus* in summer and winter (F = 5.535, 9.407, all P < 0.05) (Table 3). Fig 6 shows the survival curve of adult mosquitoes in summer and winter. The log rank test showed that the survival curves of the two groups were different, and the difference was statistically significant (P < 0.05).

## Changes in the experimental environment

During the experiment, the water temperature of the living microenvironment of the larvae was recorded using a temperature recorder, and the daily average water temperature was obtained from the hourly water temperature. The changes in water temperature during the experiment are shown in S1 and S2 Figs. Generally speaking, in winter in the field experiment, the highest temperature was 22.3°C, the lowest temperature was 10.8°C, and the average temperature was 16.9 ± 7°C. The results showed that the temperature of the field environment in Guangzhou fluctuated greatly in winter. The temperature of field environment in winter was significantly lower than that of field environment in summer and laboratory environment. In summer, the daily average water temperature was the highest, ranging from 26 to 36°C. The average daily water temperature in spring and autumn was 22.9°C. The effective accumulated temperature theory and linear regression equation $T = T_0 + K \times b$ were used to predict the eclosion threshold temperature and effective accumulated temperature of *Ae. albopictus*. The R2 of the linear regression equation model was 0.649, and the regression model was statistically significant (P < 0.001) (Table 4).

The average relative humidity of outdoor environment in summer was 79.0 ± 8.8%. Summer is a rainy season in Guangzhou; therefore, the relative humidity of the outdoor environment in summer was higher than that of the laboratory environment. The average relative humidity in winter was 73.9 ± 11.3%, the results showed that the relative humidity of Guangzhou in winter changed greatly, and the average relative humidity was still higher than that of the laboratory environment.

The light levels of the different seasons in the field environment are shown in S3 Fig. The longest photoperiod was 14 hours in summer and the shortest was 11 hours in winter. The

**Table 4. Parameters of linear regression model.**

| Independent variable | Regression coefficient | Standard error | T value | P value |
|---|---|---|---|---|
| Constant | 15.178 | 0.443 | 34.236 | <0.001 |
| 1/D | 97.185 | 4.911 | 19.789 | <0.001 |

The P values of model regression coefficients were less than 0.05, with statistical significance. The initial temperature t0 of emergence of *Ae albopictus* was 15.178°C, and the effective accumulated temperature required for emergence was 97.185°C day (°C d).

maximum light intensity in the four seasons was 1400 lux from 11:00 to 13:00, and the average light intensity in summer is 2340 lux in per 24h. The average light intensity in autumn was 1003 lux in per 24h.

## DENV-2 infection rate of *Ae. Albopictus*

The infection statuses of different tissues of *Ae. albopictus* on the 7th day after dengue virus-2 infection are shown in Table 5 and Fig 7A. The dengue virus infection rate of the midgut was relatively high under simulated field conditions (containing mosquitoes infected in summer and winter) and laboratory conditions at 7 dpi, both of which reached more than 90%. The dengue virus infection rates of the ovary and head were 56.3% and 45.7%, respectively, under laboratory conditions, and 90.2% and 91.4%, respectively, under simulated field conditions at 7 dpi. The chi squared test was used to test the infection rate of the same tissue in different experimental environments at 7 dpi. The results showed that there were significant differences in the infection rate of the ovary at 7 dpi under different experimental conditions ($\chi^2 = 672$, $P < 0.05$). The infection rate of head was also statistically different ($\chi^2 = 608$, $P < 0.05$). The infection rate of the ovary and head at 7 dpi under simulated field conditions was higher than that under laboratory conditions. At 14 dpi, as shown in Table 5 and Fig 7B, the infection rates of the ovary, midgut, and head were all above 90% under simulated field conditions, and were above 85% under laboratory conditions. The chi squared test was used to test the infection rate of the same tissue in different experimental environments, which showed that there was no significant difference in infection rate between the two groups at 14 dpi ($P > 0.05$).

Under the same experimental conditions, the infection rate of the same tissue at different time points is shown in (Fig 7C and 7D). Under laboratory conditions, the infection rates of the ovary and head at 7 dpi were 44.0% and 56.0%, respectively, and at 14 dpi were 90.7% and 85%, respectively. The chi squared test showed that the ovarian infection rate at 7 dpi was lower than that at 14 dpi ($\chi^2 = 629$, $P < 0.05$), and the head infection rate at 7 dpi was lower than that at 14 dpi ($\chi^2 = 8.608$, $P < 0.05$). Under simulated field conditions, the infection rates of the ovary, midgut, and head were all above 88%. There was no significant difference in the infection rates of the same tissue between 7 and 14 dpi ($P > 0.05$). The results showed that mosquitoes in simulated field environment were more susceptible to DENV-2 than those in the simulated laboratory environment.

The infection rates of the same tissue in different seasons are shown in (Fig 7E and 7F).

**Table 5. Rates of dengue virus infection, dissemination, potential transmission, population potential transmission by *Ae. albopictus* under different environment conditions.**

| Group | IR | DR | TD |
|---|---|---|---|
| | (%) | (%) | (%) |
| Infection group in simulated laboratory environment | 89.5 | 75.3 | 66.7 |
| Infection group in simulated field environment | | | |
| (including summer and winter) | 93.7 | 88.6 | 76.9 |
| The 7th day of Infection group in simulated | | | |
| field environment(including summer and winter) | 85.5 | 83.3 | 58.3 |
| The 14th day of Infection group in simulated | | | |
| field environment(including summer and winter) | 94.1 | 90.6 | 75.8 |
| Infection group in simulated winter field | | | |
| environment | 76.8 | 79.2 | 60.7 |
| Infection group in simulated summer field | | | |
| Environment | 90.5 | 89.0 | 73.8 |

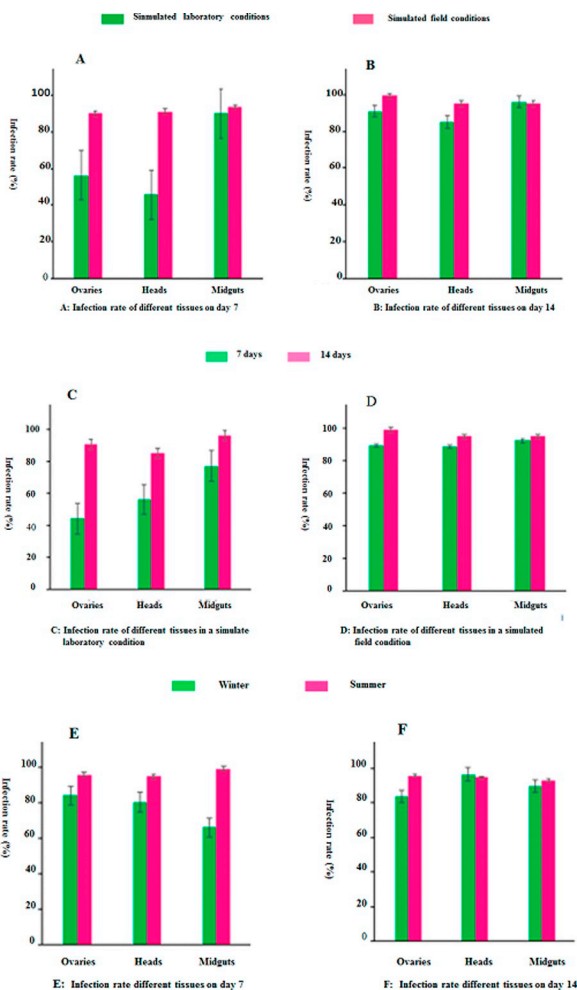

**Fig 7. DENV-2 infection rate of *Ae. albopictus*.** The experiment was divided into two groups, the simulated field environment group and the standard laboratory environment group. The simulated field environment groups was classed into simulated in summer and winter. Three factors, temperature, humidity, and light, were emphasized in the simulation of field environment in summer and winter, and the differences in the effects of these factors on mosquito infection with DENV virus were observed. (A) Infection rates of different tissues on day 7 in simulated field including summer and winter together. (B) Infection rates of different tissues on day 14 in simulated field including summer and winter together. C: Infection rate of different tissues in a simulate laboratory condition, D: Infection rate of different tissues in a simulated field condition, (E) Infection rates of different tissues on day 7 in simulated field in summer and winter. (F) Infection rates of different tissues on day 14 in simulated field in summer and winter.

The infection rates of the ovary and head at 7 dpi in summer and winter were higher than 80%. The infection rate of the midgut was 99.0% in summer, but only 66% in winter. The infection rate of the same tissue at 7 dpi was tested using the chi squared test. The infection rate of the midgut in summer was higher than that in winter ($\chi^2$ = 8.452, P < 0.05). The infection rates in winter and summer were all about 90%, and there was no significant difference in the infection rates in different seasons at 14 dpi (P > 0.05).

## Amount of dengue virus in the midgut, ovary and head of mosquito

The results of IR, DR, TR, vector susceptibility indexes of *Ae. albopictus* infected with DENV-2 at 7 and 14 dpi in winter and summer under laboratory and simulated field conditions are shown in Table 5. There was significant difference between them by One-Way ANOVA (IR: F = 253.6, p<0.001; DR: F = 253.6, p<0.001; TD: F = 253.6, p<0.001). qPCR was used to detect

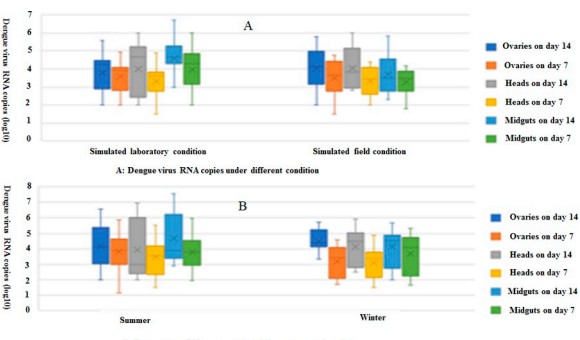

**Fig 8. Amount of dengue virus in the midgut, ovary, and head of mosquitoes under different conditions.** (A) Dengue virus RNA copies under different condition. (B) Dengue virus RNA copies under different seasonal conditions.

the dengue virus load in the midgut, head, and ovary of the mosquitoes. The average dengue virus RNA copy number (log10) in each tissue in the same season is shown in Fig 8A and Table 6. There was no significant difference in the RNA copy number of mosquitoes infected with dengue virus in the ovaries, head, and midgut tissues at 7 and 14 dpi ($P > 0.05$). The results of variance tests showed that the mean copy number of virus RNA in the midgut at 7 dpi in summer was higher than that at 7 dpi in winter ($F = 14.459$, $P = 0.01$). The mean copy number of virus RNA in the ovary at 7 dpi in summer was also higher than that at 7 dpi in winter ($F = 8.656$, $P < 0.001$); however, there was no significant difference in the number of virus copies at other time points ($P > 0.05$) (Fig 8B, S5 Table).

As shown in S4 Fig, for *Dicer-2*, *Rel-1*, and the β-actin gene, the correlation coefficients R2 of the standard curve were 0.9996, 0.9973, and 0.9986, respectively.

## Effects of immune-associated genes on vector competence of *Ae. albopictus*

For different days in different seasons, the DENV-2 loads and the expression levels of immune-associated genes in whole infected mosquitoes were determined by qRT-PCR

**Table 6. Average copy number of dengue virus RNA in midgut, head and ovary of *Ae. albopictus* infected under different conditions.**

| Group | Midgut | Ovary | Head |
|---|---|---|---|
| The 7th day of infection group in simulated laboratory environment | 3.96 | 3.32 | 3.57 |
| The 14th day of infection group in simulated laboratory environment | 4.61 | 3.99 | 3.77 |
| The 7th day of Infection group in simulated field environment (including summer and winter) | 3.27 | 3.30 | 3.50 |
| The 14th day of Infection group in simulated field environment (including summer and winter) | 3.70 | 4.06 | 4.02 |
| The 7th day of Infection group in simulated summer field environment | 3.81 | 3.52 | 3.85 |
| The 14th day of Infection group in simulated summer field environment | 4.69 | 3.96 | 4.21 |
| The 7th day of Infection group in simulated winter field environment | 3.69 | 3.10 | 3.20 |
| The 14th day of Infection group in simulated winter field environment | 4.14 | 4.13 | 4.49 |

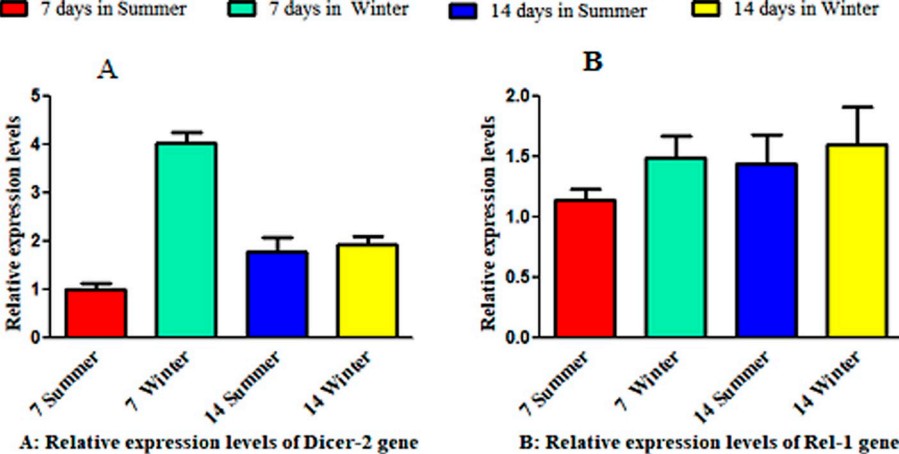

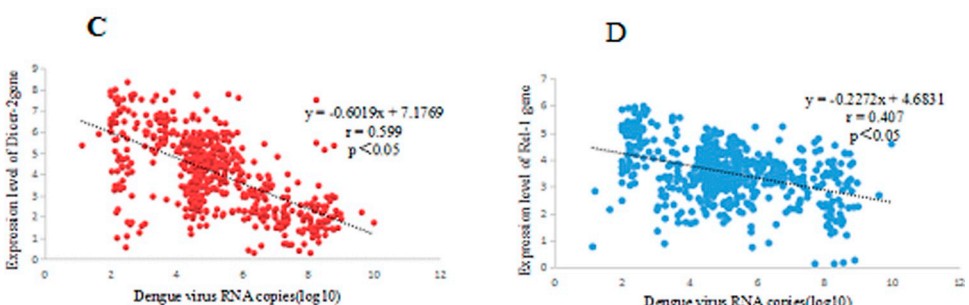

**Fig 9. Correlations between DENV-2 loads and the expression levels of immune- associated genes [*Dicer-2, Rel2*] in the midguts of *Ae. albopictus* at 7 and 14 days after dengue virus infection in different seasons.** (A) *Dicer-2* gene expression levels. (B) *Rel-1* gene expression levels. (C) Correlations between DENV-2 loads and *Dicer-2* gene expression levels. (D) Correlations between DENV-2 loads and *Rel-1* gene expression levels.

combined with the comparative CT value method. The expression of *Dicer2* at 7 dpi winter was 4.071 times that at 7 dpi in summer (P < 0.05; Fig 9A, S5 Table). There was no significant difference in *Dicer-2* gene expression at 14 dpi between summer and winter (P > 0.05). The relative expression levels of *Rel-1* at 7 dpi in winter, 14 dpi in summer, and 14 dpi in winter were 1.484, 1.440, and 1.595 times higher, respectively, than that at 7 dpi in summer. The results showed that there was no significant difference in *Rel-1* expression at 7 dpi and 14 dpi between summer and winter (both P > 0.05; Fig 9B, S6 Table). As shown in S4 Fig, for *Dicer-2*, *Rel*-1, and the β-actin gene, the correlation coefficients R2 of the standard curve were 0.9996, 0.9973, and 0.9986, respectively.

The correlation between dengue virus load and immune gene expression in the midgut of *Ae albopictus* was analyzed, and the results are shown in Fig 9. The results of correlation analysis between dengue virus load and the expression of immune gene *Dicer-2* showed a Pearson correlation coefficient r = −0.599 (P < 0.05), and the absolute value of the correlation coefficient r was in the range of 0.5<|r|<0.8 (Fig 9C). There was a moderate but significant correlation between dengue virus load and the expression of immune gene *Dicer-2*. Therefore, with the increase of dengue virus load, the expression of *Dicer-2* decreased. The results of

correlation analysis between dengue virus load and immune gene *Rel-1* expression showed a Pearson correlation coefficient r = −0.407 and P < 0.05, and the absolute value of correlation coefficient r was in the range of 0.3<|r|<0.5 (Fig 9D).

There was a low but significant correlation between dengue virus load and immune gene *Rel-1* expression.

## Fluctuation of daily mean temperature and humidity

During the period of mosquito infection, the daily average temperature and humidity fluctuations in the field are shown in S5 Fig. In summer, the maximum daily average temperature was 31.9˚C, the minimum temperature was 26.4˚C, the maximum daily average relative humidity was 96.0%, and the minimum was 76.7%. In winter, the highest daily average temperature was 21.4˚C, the lowest was 10.6˚C, the highest daily average relative humidity was 86.0%, and the lowest was 67.0%.

The changes of temperature and humidity during mosquito infection are shown in S6 Fig. Generally speaking, during the whole experiment, Guangzhou had obvious characteristics of a hot and humid summer. In summer, the average temperature in the field was the highest at 14:00, reaching 32.9˚C, the lowest average temperature was 26.4˚C at 6 a.m. The average temperature fluctuated from 26.4 to 32.9˚C, the relative humidity fluctuated from 55.4% to 100%, with an average 84%. In winter, the average temperature in the field was the highest at 14:00, reaching 22˚C, the lowest average temperature was 15˚C at 1 a.m. During the whole experiment, the average temperature fluctuated from 15 to 22˚C, the relative humidity fluctuated from 71% to 95%, and the average humidity was 82%.

## Discussion

Dengue fever is an important mosquito borne disease transmitted by *Ae. albopictus*. Climate change, ecological environment change, globalization, and other factors have an important impact on the occurrence and prevalence of the disease [69]. In China, Guangzhou city in Guangdong Province has the high incidence dengue fever. Guangzhou is located in the southeast coastal area of China, with developed transportation and economy, frequent communication with people at home and abroad, and a subtropical monsoon climate. Therefore, it is meaningful to explore the influence of seasonal climate change on the growth and development of *Ae. albopictus* by simulating the field environment in Guangzhou.

Climatic factors, especially the temperature difference between different seasons, play an important role in the growth and decline of the density of mosquito populations. The prevalence of dengue fever is related to the density of mosquito vectors, and the peaks of dengue fever and mosquito density are in summer and autumn, with high temperature and rainy weather [15]. In the field, the pupation rate and eclosion rate of the Guangzhou strain and laboratory strain were above 80% in the four seasons, and there was no significant difference in the pupation or eclosion rate in the different seasons. In the same season, there was no significant difference between the pupation rate and emergence rate of *Ae. albopictus* under the optimum temperature in the laboratory and in the field. The results showed that temperature had no significant effect on the pupation rate and eclosion rate of larvae in Guangzhou, and the eclosion rate in the four seasons was comparable to that in the most suitable living environment (the laboratory environment), This is consistent with our previous report [13]. In this study, the average temperature in winter was 18.2˚C, and the pupation rate of Guangzhou was 96%. The average temperatures in summer and autumn were 28.8 and 22.9˚C respectively, and the pupation rates were 96% and 95% respectively. Thus, the pupation rate remains at a high level under different temperature conditions in the field.

Interestingly, during the experiment, the average temperatures in spring and autumn were 22.98˚C and 22.94˚C respectively, and the fluctuation trend of daily temperature was basically the same, while the average duration of larval development to eclosion was 15 days and 9 days respectively; the difference was statistically significant. The average temperature difference between summer and autumn is about 5˚C, and the time from development to emergence was 9 days. This showed that when the environmental temperature was between 20 and 30˚C, the temperature did not have a simple linear negative correlation with the development period of *Ae albopictus* larvae, which was different from the conclusion that the higher the temperature, the shorter the developmental period when different temperature gradients are set in the laboratory [70, 71]. In the field environment, the fluctuation of daily temperature and illumination in different seasons are different from the artificial conditions in the laboratory, which have different effects on the development and reproduction of *Ae. albopictus*. The duration of mosquito development in summer and autumn is the shortest, which might be related to the seasonal rhythm of mosquitoes. The mosquito density was consistent with the epidemic trend of dengue in Guangzhou in summer and autumn.

In another simulation experiment in this study, the growth and development of *Ae. albopictus* in summer and winter were significantly different. The hatching rate of larvae in summer was higher than that in winter. The incubation time of *Ae. albopictus* was shorter in summer and longer in winter, and the hatching rate was relatively low in winter. This might be caused by the low temperature in winter, or the preference of *Ae. albopictus* to overwinter in the form of eggs. This is similar to the results reported by Zheng et al. [14] and Yang et al. [13]. The results showed that *Aedes* mainly overwinters as eggs, but also in the form of larvae and residual adult mosquitoes, which is termed semi-overwintering. Seasonal climate change will significantly affect the hatching rate of *Ae. albopictus* larvae. A lower temperature might reduce the reproductive capacity of female adults. In winter, the number of eggs laid by mosquitoes in the outdoor environment was quite different from that of adults in other environmental conditions. In summer, the average oviposition of mosquitoes in the field was 4123.333; in the winter group, two groups of mosquitoes in field group did not lay eggs, and the average number of eggs laid in the other two groups was 140.5. We found that in the field experiments in winter, some *Ae. albopictus* larvae might enter diapause when they develop to the fourth instar larva. This was consistent with the conclusions of Yang et al. [8, 13]. The survival time of adult *Ae. albopictus* in summer was longer than that in winter. The longest survival time of adult *Ae. albopictus* in summer was 54 days. More importantly, the survival time of females was longer than that of males. The average survival time of female mosquitoes was 16.87 days, and that of male mosquitoes was 12.09 days. We observed that adult mosquitoes had a long life span in the field in summer, which can fully meet the EIP time required for mosquito to transmit dengue virus, which is consistent with the peak incidence of dengue fever in Guangzhou in summer [72]. In winter, the longest survival time of adult mosquitoes in the field environment was 36 days, and the average survival time was 10.08 ± 10.60 d (mean ± SD). *Ae. albopictus* could still develop in winter in Guangzhou. However, under the influence of lower temperature, the number of eggs laid by female *Ae. albopictus* was less, the hatching rate and pupation rate were lower, and the larval development time was prolonged. This suggested that *Ae. albopictus* still has the potential to transmit dengue virus in winter in Guangzhou, and there is a risk of a dengue fever epidemic.

Vector ability is the ability of arthropod vectors to be infected by pathogens, propagate the pathogens in their bodies, and transmit the viruses to other hosts. To date, *Ae. albopictus* is the only vector of dengue virus in Guangzhou. It is necessary to evaluate the vector ability of *Ae. albopictus* for dengue fever in Guangzhou. To evaluate the risk of dengue virus transmission in mosquito population accurately, it is necessary to incorporate local temperature into the

experiment of vector capacity of *Ae. albopictus*. Many studies in the laboratory set a constant temperature to study the time required for dengue virus to propagate in mosquitoes to be infectious. The rates of infection, dissemination, population transmission, and DENV-2 copies at 28˚C were higher than those at 23˚C at any time point. At 32˚C, the extrinsic incubation period (EIP) for DENV-2 in *Ae. albopictus* was only 5 dpi, and the vector competence was the highest among all the temperatures. Compared with 28˚C, at 28–23–18˚C, the positive rate and the amount of DENV-2 in the salivary glands were significantly lower [56]. Therefore, temperature is an important factor affecting the vector competence of *Ae. albopictus* for DENV-2. Within the suitable temperature range, the replication of DENV-2 in *Ae. albopictus* accelerated, and the EIP was shorter with higher temperature [56]. Xiao et al. found that the positive virus detection time of the salivary glands of mosquitoes reared at 21, 26, 31, and 36˚C were 10, 7, 4, and 4 days, respectively. With the increase in temperature, the EIP of DENV-2 in *Ae. albopictus* gradually shortened [73]. However, some studies found that the EIP of DENV-2 virus in *Ae albopictus* was different from that under constant temperature. Carrington et al. found that the EIP of dengue virus in *Aedes aegypti in vivo* and *in vitro* was shortened when the temperature was low and fluctuated greatly [74]. The temperature fluctuation reflects the mosquito living environment more truly than constant temperature conditions. The best way to estimate the risk of dengue fever outbreak is to measure whether mosquitoes are infected with dengue virus while simulating environmental conditions at mosquito collection sites [75]. In this study, we used a temperature and humidity recorder to record the hourly temperature and humidity of mosquitoes living in the field, and simulated the hourly temperature and humidity changes in the field environment in an artificial climate box, which was more detailed than the temperature fluctuation conditions set by Brustolin et al. (daytime temperature 26˚C, nighttime temperature 22˚C [75]. In the present study, the temperature fluctuated from, the relative humidity fluctuated from 55.4 to 100%, and the average humidity was 84% in summer. By contrast, the temperature fluctuated from 8.5 to 22˚C, the relative humidity fluctuated from 71 to 95%, and the average humidity was 82% in winter.

The dengue virus infection rate of midgut was higher under simulated field and laboratory conditions, both of which were more than 90% at 7 dpi. The infection rates of the ovary and head were 56.3% and 45.7%, respectively, under laboratory conditions, and 90.2% and 91.4%, respectively, under simulated field conditions. The infection rate of the ovary and head at 7 dpi in the simulated field was higher than that in the laboratory. The results showed that the Asia tiger mosquitoes in simulated field environment were more susceptible to DENV-2 than those in simulated laboratory environment. The hourly average temperature of the simulated field summer environment fluctuated between 26.4 and 32.9˚C. The average temperature of the whole experimental period was 28.7˚C, which is close to that of the simulated laboratory environment (28˚C). The results showed that the infection rate of the ovary and head of *Ae. albopictus* was higher in the simulated field conditions than in the laboratory at 7 dpi. This might indicate that the average temperature is close, and the temperature fluctuation in the living microenvironment of *Ae. albopictus* will shorten the time of dengue virus infection of the ovary and head, i.e., shorten the EIP. Thus the EIP in the natural environment might be shorter than that in laboratory.

The results showed that there was no significant difference in the infection rate and viral load of *Ae. albopictus* in the midgut, ovary, and head between 7 and 14 dpi in summer, and the infection rate and viral load remained at a high level, indicating that dengue virus had infected the above tissues before day 7 and the viral load had reached its peak. It was suggested that the high temperature and humidity in summer shortened the EIP of dengue virus. During the large-scale outbreak of dengue fever in Guangzhou China in 2014, Guangdong Meteorological service data showed that the monthly average temperature from June to September was 0.1–

1.3˚C higher than that of previous years. The rainfall was 63% higher than usual [65]. The results have important reference value in dengue vector control and epidemic control. The results showed that ability of mosquitoes to transmit dengue in the field was stronger than that in the other experimental groups.

In addition, we found that the average copy number of dengue virus RNA in the midgut and ovary of mosquitoes at 7 dpi in summer was higher than that at 7 dpi in winter. We speculated that the mechanism might be related to the regulation of immune-related genes in different seasonal environments. Our results showed that the expression of *Dicer2* at 7 dpi in winter was significantly higher than that at 7 dpi in summer, and the difference was statistically significant. The expression of *Rel-1* at 7 dpi in winter, 14 dpi summer, and 14 dpi winter was 1.484, 1.440, and 1.595 times higher than that at 7 dpi in summer. The results showed that the expression levels of *Dicer2* and *Rel-1* correlated negatively with the viral load in mosquitoes, suggesting that the downregulated expression of these two genes was conducive to viral reproduction. Adelman et al. reported that cooler temperatures destabilize RNA interference and increase the susceptibility of disease vector mosquitoes to viral infection. They utilized transgenic "sensor" strains of *Ae. aegypti* to examine the role of temperature on RNA silencing. These "sensor" strains express the enhanced green fluorescent protein (EGFP) only when RNAi was inhibited; for example, after knockdown of the effector genes encoding proteins *Dicer-2* or *Argonaute-2*. They observed an increase in EGFP expression in transgenic sensor mosquitoes reared at 18˚C as compared with those at 28˚C [76]. These results are consistent with the observation that expression of *Dicer2* was high at 7 dpi in winter in this study. Wimalasiri-Yapa et al., reported that temperature modulates immune gene expression in mosquitoes during arbovirus infection. They exposed Chikungunya virus (CHIKV)-infected mosquitoes to 18, 28 and 32˚C, and found that higher temperature correlated with higher virus levels, particularly at 3 dpi, whereas lower temperature resulted in reduced virus levels. RNA sequencing analysis indicated significantly altered gene expression levels during CHIKV infection. The highest number of significantly differentially expressed genes was observed at 28˚C, with a more muted effect at other temperatures. At the higher temperature, the expression of many classical immune genes, including *Dicer2*, was not substantially altered in response to CHIKV [77]. These results were consistent with our results.

There is a defect in our study: the use of head samples to determine potential transmission is not standard enough compared with the use of mosquitoes salivary gland samples. In addition, in winter, the transmission of mosquito virus is lacking of animal model verification test of infected mosquitoes biting mice, so there are some defects in this study.

In this study, it was found that *Ae. albopictus* could still develop in winter in Guangzhou.

The results of the present study suggest that *Ae. albopictus* still has the ability to transmit dengue virus in winter, and thus good epidemic prevention and control measures are necessary in Guangzhou.

## Supporting information

**S1 Fig. Temperature changes during the experiments in the wild.**
(TIF)

**S2 Fig. Temperature and humidity changes during experiments in the wild in summer and winter.**
(TIF)

**S3 Fig. Changes in daily light intensity during the experiments in the wild.**
(TIF)

**S4 Fig. The standard curve of Dicer-2, Rel-1, and β-actin expression.**
(TIF)

**S5 Fig. Changes in the relative temperature and humidity during the experiment in different environments.** A: Temperature changes during the experiment in different environments. B: Variation of relative humidity in different environments.
(TIF)

**S6 Fig. Changes in the daily mean temperature and humidity in the field during the experiment.** A: Daily average temperature and relative humidity. B: Average temperature and relative humidity per hour.
(TIF)

**S1 Table. Hatching results of *Ae. albopictus* eggs under different environmental conditions.**
(DOCX)

**S2 Table. Pupation rate and pupation time of *Ae. albopictus* larvae under different environmental conditions.**
(DOCX)

**S3 Table. Eclosion results of *Ae. albopictus* pupae under different environmental conditions.**
(DOCX)

**S4 Table. Oviposition of adult *Ae. albopictus* under different environmental conditions.**
(DOCX)

**S5 Table. Average air temperature and relative humidity during the experiment.**
(DOCX)

**S6 Table. The expression of Dicer-2 and Rel-1 genes in the midguts of Aedes albopictus in 7 and 14 days-post-infection (dpi) in summer and winter experiments.**
(DOCX)

## Author Contributions

**Conceptualization:** Xueli Zheng.

**Data curation:** Xueli Zheng.

**Formal analysis:** Xueli Zheng.

**Funding acquisition:** Xueli Zheng.

**Investigation:** Xueli Zheng.

**Methodology:** Shanshan Wu, Yulan He, Yong Wei, Peiyang Fan, Weigui Ni, Xueli Zheng.

**Project administration:** Xueli Zheng.

**Resources:** Xueli Zheng.

**Supervision:** Xueli Zheng.

**Validation:** Shanshan Wu, Yulan He, Yong Wei, Peiyang Fan, Weigui Ni, Xueli Zheng.

**Visualization:** Xueli Zheng.

**Writing – original draft:** Xueli Zheng.

**Writing – review & editing:** Daibin Zhong, Guofa Zhou, Xueli Zheng.

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
