## [Decision Letter · Decision Letter 0]

13 Jan 2022

PONE-D-21-26984Effects of Guangzhou seasonal climate change on the development of Aedes albopictus and its susceptibility to DENV-2PLOS ONE

Dear Dr. Zheng,

Thank you for submitting your manuscript to PLOS ONE. After careful consideration, we feel that it has merit but does not fully meet PLOS ONE’s publication criteria as it currently stands. Therefore, we invite you to submit a revised version of the manuscript that addresses the points raised during the review process.

ACADEMIC EDITOR: Dear Dr. Zheng

I received the review of two independent reviewers. Both required additional information to strengthen the manuscript you submitted. Special concerns were raised in missing information from methods and results sections, which strongly affected the manuscript clarity in several ways. One of the reviewers sent attached a PDF file with additional comments.

We look forward to receiving your revised manuscript.

Kind regards,

Rafael Maciel-de-Freitas

Academic Editor

PLOS ONE

Journal Requirements:

Whilst you may use any professional scientific editing service of your choice, PLOS has partnered with both American Journal Experts (AJE) and Editage to provide discounted services to PLOS authors. Both organizations have experience helping authors meet PLOS guidelines and can provide language editing, translation, manuscript formatting, and figure formatting to ensure your manuscript meets our submission guidelines. To take advantage of our partnership with AJE, visit the AJE website (http://aje.com/go/plos) for a 15% discount off AJE services. To take advantage of our partnership with Editage, visit the Editage website (www.editage.com) and enter referral code PLOSEDIT for a 15% discount off Editage services.  If the PLOS editorial team finds any language issues in text that either AJE or Editage has edited, the service provider will re-edit the text for free.

This work was supported by the National Natural Science Foundation of China (No. 31630011), the Natural Science Foundation of Guangdong Province (No. 2017A030313625), and the Science and Technology Planning Project of Guangzhou (No. 201804020084).

OK

NO

7. Please upload a copy of Figure S3, to which you refer in your text on page 46. If the figure is no longer to be included as part of the submission please remove all reference to it within the text.

Reviewers' comments:

Reviewer's Responses to Questions

**Comments to the Author**

1. Is the manuscript technically sound, and do the data support the conclusions?

Reviewer #1: No

Reviewer #2: Yes

2. Has the statistical analysis been performed appropriately and rigorously? 

Reviewer #1: No

Reviewer #2: Yes

3. Have the authors made all data underlying the findings in their manuscript fully available?

Reviewer #1: No

Reviewer #2: Yes

4. Is the manuscript presented in an intelligible fashion and written in standard English?

Reviewer #1: No

Reviewer #2: Yes

5. Review Comments to the Author

Reviewer #1: The article has potential. But the introduction is extensive and misses the main subject. The hypothesis and the objectives are not clear. The Methods and Results of the statistical tests are not clear. The statistical results presented in the table and figures are not making sense, as well as the groups compared. The four stations appear in the group "environment" and "laboratory", but this description of the result is incompatible with the methodology.... The article has basic problems for answer the objective, such as the description of temperatures and humidity in each group compared.

Reviewer #2: Suggestions to improve clarity.

Methods.

- Line 193: “Larvae were brought back to semi-field setting and reared in microcosms where life

table experiments were conducted”. Please, provide a more detailed description of the “microcosm” used to rear the mosquitoes.

- Line 196: “F3 eggs were used for first round of life-table experiments for three reasons, to allow for get enough eggs within one day, to allow for field mosquitoes to adapt the new environment and mouse blood.”. Please, restructure this sentence to improve clarity.

- Line 203: Is it possible to use another title for this section? It is not clear to me what the authors mean by “Semi-Life scale experimental methods”.

- Line 250. It is not clear to me the section “Expanded culture of experimental mosquitoes”, including the title. Particularly the last sentence “In July 2019, eggs of Ae. albopictus from Guangzhou were released in both the field and laboratory environments.”.

- Line 275. “the mosquito was placed a glass dish on ice”. Please, correct the sentence.

- 288 to 292. Overall I think the description of study design is not straightforward. The section described from line 288 to 292 is an example.

- Line 317. “cDNA was synthesized using a DENV-2-specific primer (5'-TGGTCTTTCCCAGCGTCAAT-3')”. Is this correct? Only one primer was used? Please, clarity.

Results:

- Line 461. Figure 2. Study design was not clear to me, especially what “control” means. Authors could use figure legends to clarify.

- Line 525. Figure 5. Please, add number of mosquitos analyzed in figure legends, so readers can at least average egg/mosquito without having to search for this information in the text. Please, define in figure legends what authors mean by “control”.

- Line 655. I could not find figure 9C and 9D.

Discussion:

- Line 858. Please, restructure sentence to improve clarity.

6. PLOS authors have the option to publish the peer review history of their article (what does this mean?). If published, this will include your full peer review and any attached files.

Reviewer #1: No

Reviewer #2: No

---

## [Author Response · Author response to Decision Letter 0]

8 Mar 2022

Department of Pathogen Biology

School of Public Health

Southern Medical University

1838 Guangzhou North Av.

Guangzhou 510515, China

Tel: +86-20-6164-8651

E-mail: zhengxueli2001@126.com

On March 3, 2022 

 Dear PLOSE ONE Editor:

Thank you very much for your email dated on 2020/14/1 with comments attached for our manuscript ((PONE-D-21-26984). We have now made a point-to-point response to all reviewers and Editor (Rafael Maciel-de-Freitas Academic Editor) comments. All page and line numbers described below refer to the revised version. 

All contributing authors have reviewed and approved this revised version. The material has not been and will not be offered elsewhere for publication. Please let me know if further editorial or clarifications need to be made. The authors declare that they have no competing interests.

This work was supported by the Natural Science Foundation of China (No. 31630011), the Natural Science Foundation of Guangdong Province (No. 2017A030313625), and the Science and Technology Planning Project of Guangzhou (No. 201804020084). The funders had no role in study design, data collection and analysis, decision to publish, or preparation of the manuscript.

Thank you very much for your assistance with this paper.

Yours sincerely,

Xueli Zheng, Ph.D.

Professor

---

## [Editor Report · Decision Letter 1]

15 Mar 2022

Effects of Guangzhou seasonal climate change on the development of Aedes albopictus and its susceptibility to DENV-2

PONE-D-21-26984R1

Dear Dr. Zheng,

We’re pleased to inform you that your manuscript has been judged scientifically suitable for publication and will be formally accepted for publication once it meets all outstanding technical requirements.

Kind regards,

Rafael Maciel-de-Freitas

Academic Editor

PLOS ONE

---

## [Editor Report · Acceptance letter]

24 Mar 2022

PONE-D-21-26984R1 

Effects of Guangzhou seasonal climate change on the development of *Aedes albopictus* and its susceptibility to DENV-2 

Dear Dr. Zheng:

I'm pleased to inform you that your manuscript has been deemed suitable for publication in PLOS ONE. Congratulations! Your manuscript is now with our production department. 

Kind regards, 

on behalf of

Dr. Rafael Maciel-de-Freitas 

Academic Editor

PLOS ONE